# Lessons from Contextual Bandit Learning in a Customer Support Bot

Nikos Karampatziakis [1]   Sebastian Kochman [1]   Jade Huang [1]   Paul Mineiro [2]   Kathy Osborne [3]   Weizhu Chen [1]

## Abstract

In this work, we describe practical lessons we have learned from successfully using contextual bandits (CBs) to improve key business metrics of the Microsoft Virtual Agent for customer support. While our current use cases focus on single step reinforcement learning (RL) and mostly in the domain of natural language processing and information retrieval we believe many of our findings are generally applicable. Through this article, we highlight certain issues that RL practitioners may encounter in similar types of applications as well as offer practical solutions to these challenges.

## 1. Introduction

Many real world systems operate in a partial information setting, meaning that they never observe what their users think about the actions they did not take. In recommendation systems such as those for news (Li et al., 2010) and video (Schnabel et al., 2016; Chen et al., 2019), the industry standard has been recently switching to contextual bandits (CB, Langford & Zhang, 2007), a simplified reinforcement learning paradigm which requires exploration but does not require dealing with credit assignment. The main challenge in many of these systems is that unlike RL with a simulator, it is not possible to observe how good other actions would have been if they had been tried in the same situation.

In this paper we present the lessons we have learned by applying single step RL in real-world scenarios in the Microsoft Virtual Agent, a conversational system for customer support. The rest of the paper is organized as follows. In Section 2 we present two real-world scenarios in which we have applied RL and discuss gains we have achieved so far from successful RL policies. In Section 3 we describe lessons we have learned from our experience with these scenarios. In Section 4 we review some systems that can help with applying RL to real-world problems.

[1]Microsoft Dynamics 365 AI [2]Microsoft Research [3]Microsoft. Correspondence to: <nikosk@microsoft.com>.

*Reinforcement Learning for Real Life (RL4RealLife) Workshop in the 36th International Conference on Machine Learning*, Long Beach, California, USA, 2019. Copyright 2019 by the author(s).

## 2. Case studies

One of our partner teams operates the Microsoft Virtual Agent – an interactive dialogue system providing first-line customer support for many Microsoft products. The bot can be accessed via multiple channels – most commonly through the Microsoft support website [1] or the Get Help app which comes with Windows 10.

RL is a good fit for this domain for multiple reasons:

1. Diversity of user intents and a changing environment (new products, updates causing new issues as well as new or updated support content) make the problem challenging for approaches that rely solely on editorial labels and supervised learning.

2. Microsoft products (including Xbox, Office, Windows, Skype) have a large customer base, hence the bot's incoming traffic is also significant, making RL potentially feasible.

3. We have access to multiple reward signals including click behavior, escalation of the issue to a human agent, and responses to survey questions such as "Did this solve your problem?". Moreover, some of the actual business metrics are very close to some of these easily measurable quantities. On the other hand we have found it challenging to translate traditional supervised learning metrics (e.g. accuracy or $F_1$ score) into business metrics. RL provides a more direct path for optimizing the final metrics.

At the same time, the application poses interesting challenges to the use of RL. Goal-oriented dialogue systems need to not only understand the user's original intent but also be able to carry the state of the dialogue and guide the user towards their goal. While our long-term plan is to be able to have a single RL agent in charge of the whole conversation, the requirements of such an endeavor, both in terms of number of samples needed to learn non-trivial policies and in terms of engineering effort, are substantial. Therefore, we started by applying RL to the virtual agent in isolated components first, ignoring issues of credit assignment and thus working in the setting of contextual bandits.

[1]https://support.microsoft.com/en-us/contactus

Compared to most literature on CBs, our action spaces are typically more complicated. They could be a combinatorial set, such as a slate, and the candidate base actions are not a fixed set but can vary based on the user's statement. Finally, the actions themselves have features such as title, type of content, etc.

In the following subsection, we describe two concrete scenarios where we have applied real-world RL in the Microsoft Virtual Agent.

## 2.1. Intent Disambiguation

The domain of customer support, especially for a large company such as Microsoft, is complex, with a significant number of intents. Our intent disambiguation policy is tasked with deciding when the user's query is clear enough to directly trigger a solution, such as a multi-step troubleshooting dialogue, or to ask a clarification question.

The inputs to this policy are the statement of the issue by the user (the query), the user context, and a list of candidates. The query can range from very short, including just keywords, to long and complex sentences or even paragraphs. The user context includes information like the user's operating system and its version (e.g. Office products work on Mac, iOS, and Android). The list of candidates is a collection of pre-authored dialogue intents or solutions related to the user's query pulled from the Web. Retrieval of these candidates is currently performed by several strategies. One of them includes a deep learning model similar to the one described in (Huang et al., 2018). Another uses Bing Custom Search for customized document retrieval. These retrieval components are currently out-of-scope of RL-based optimization so we will focus on the policy that operates with a small list of retrieved candidates.

Given the input, the policy can take one of the following actions:

- Directly trigger a single intent or a solution: this may mean starting a troubleshooting dialogue or displaying a rich text solution.

- Ask a yes/no question: "Here's what I think you are asking about: . . . . Is that correct?".

- Ask a multiple-choice question: "Which one did you mean?", followed by titles of two to four intents as well as option "None of the above"

- Give up: "I'm sorry, I didn't understand. It helps me when you name the product and briefly describe the issue."

Figure 1 presents an example of a multi-choice question action taken by the disambiguation policy.

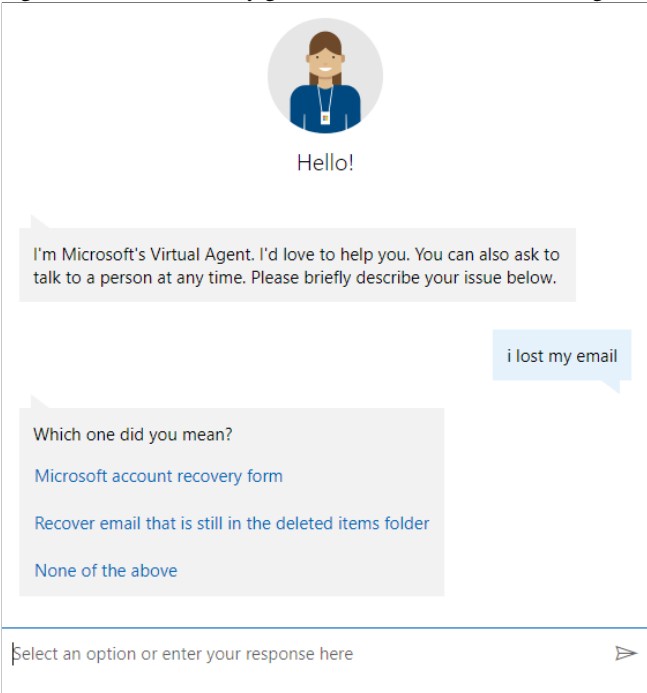

*Figure 1.* An automatically generated intent clarification dialog.

There are multiple reward signals that we currently use:

- Click: if a disambiguation question is asked, an option selected by the user is recorded. This signal is censored for the direct trigger and give-up actions. While useful, it is not a metric important to the business, so we do not optimize for it alone.

- Problem resolution: after providing a final answer to the user, the bot asks whether the solution was helpful. The user may respond "yes", "no", or decline to answer. This response becomes our main reward signal.

- Escalation: the user can decide at any time to talk to a human agent. This negative reward signal is directly related to the actual cost of running a call center. The metric is monitored but it is not being optimized.

Some reward signals may be delayed. A long conversation may occur between the disambiguation policy's action and the point at which we ask the user if the proposed solution addressed their problem. However, currently we ignore this fact and attribute the reward to the policy's action. All the actions taken by the virtual agent which are not part of the disambiguation policy are simply treated as part of the environment outside of the RL agent's control. This simplified view of the problem allows us to make progress in applying RL to a complex enterprise system while still

*Figure 2.* Contextual recommendations for the "Printers" settings page.

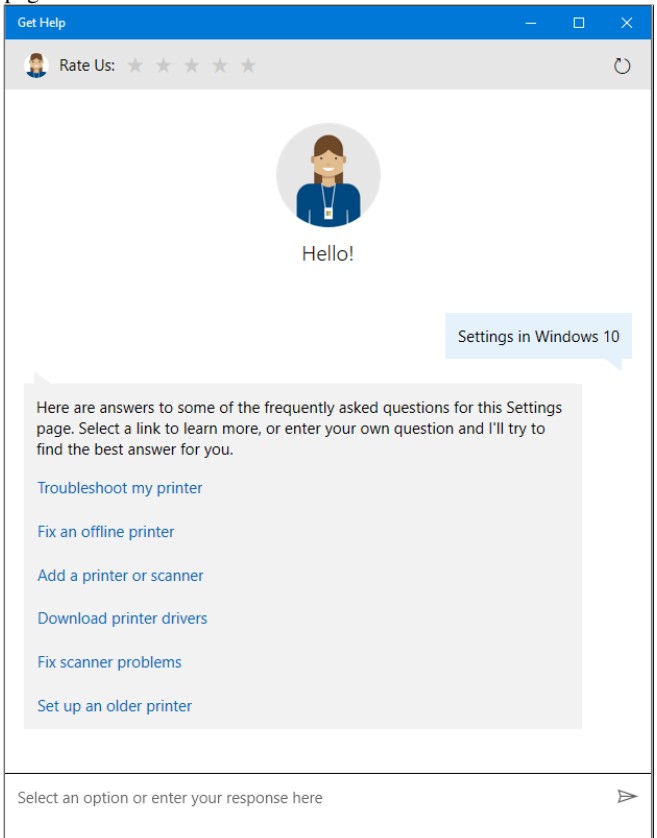

collecting data that could be used to bootstrap a multi-step policy.

## 2.2. Contextual Recommendations

"Settings" is a desktop application in Windows 10, where a user can click "Get help" from any settings page (e.g. "Bluetooth", "Display" etc.) and interact with the Microsoft Virtual Agent. We are currently experimenting with using RL for contextual recommendations, i.e. recommending a list of solutions before the user types anything, simply based on context information sent by the app.

Several reasons motivate us to leverage a model to contextually suggest recommendations. First, we do not have enough editorial bandwidth to hand-specify dialogues for each of the pages. Second, a large amount of traffic flows through the app, which suggests that data-driven techniques could do well here. Third, we log a rich user context that includes data such as from which page the user is clicking "Get help", the device network type (wired, wifi), battery status (charging, discharging), and manufacturer, which can help with suggesting relevant solutions in certain situations.

Inputs to the model are a subset of the user context and a list

of candidate solutions. The model is trained to maximize similarity between a given user context and content that corresponds to queries the users type.

Given the inputs, the model can take one of the following actions:

1. Suggest up to six solutions out of about 3000 candidates. A filtering step, using the same model, first reduces the number of candidates to up to 18[2], before performing exploration.

2. Choose to fallback to default behavior of recommending six fixed options chosen by editors in the case the model does not have enough confidence.

Reward signals are similar to the ones described in section 2.1.

## 3. Lessons

This section highlights certain factors of a real-world RL system which we found crucial to the final outcome of the project. They are presented in the form of concise lessons which we believe RL practitioners may find useful.

### 3.1. Use and Extend Existing Systems

Our design borrows many ideas behind the Decision Service (DS Agarwal et al., 2016) system. In particular, unlike many simulator-backed RL applications, which train with on-policy data, we optimize using off-policy data. Updates to the policy under which we act and collect data (the *logging* policy) happen in a regular schedule such as once a day. In between these updates, the logging policy is fixed. Collected data (including the features, the chosen action, the probability of the chosen action, and the observed reward) from the current and past logging policies are used to train and evaluate the next iteration of the logging policy. We deliberately do not prescribe exact criteria for how often to train, how much historical data to use, and when to replace the current champion policy. These should be decided on the basis of each application.

Our work has been greatly simplified by using parts of the DS system. We had to do very little work to get logging and exploration working inside our codebase. This allowed us to focus on extending the parts that Decision Service only covered for linear models. We therefore focused on training, evaluation, and deployment of deep learning models, in either automated or ad-hoc fashion. We have extended the DS with the following capabilities:

---

[2]Six solutions must be shown and 18 is the minimum number needed so that after executing business rules, in the worst case, six solutions will be left.

**Ability to use a custom model**: In many application domains, it is common to have an established architecture for state-of-the-art results given supervised data. Examples include ResNets (He et al., 2016) in object recognition and LSTMs (Hochreiter & Schmidhuber, 1997) in handwriting recognition. It is therefore important to allow domain experts to seamlessly use their own model architecture for their policy. Recent efforts such as Core ML (Apple) and ONNX (ONNX) have facilitated the standardization of inference APIs. In our applications we assume the model can be expressed as an ONNX graph.

**Ability to log different kinds of reward signals**: In many cases there are multiple reward signals that could be used in lieu of the actual reward. For example, in recommendation systems, clicks and dwell times have long been used as implicit ratings. But the exact way to combine these into a final reward might require many iterations of reward shaping/engineering. It is therefore important to be able to experiment with different definitions offline by having access to all available reward signals.

**Support for deployment with guard rails**: We maintain a dashboard with counterfactual evaluation results for different rewards and recently trained models. Deployment can be automatic, depending on prespecified conditions, or manual based on a human operator's decision.

## 3.2. Pay Attention to Effective Sample Size

Since we are typically learning from off-policy data, many ideas and diagnostics developed in the importance sampling literature (Owen, 2013) can be used to help debug computational and statistical problems. One popular diagnostic is the *effective sample size* which can be thought of as the number of full-information examples we can extract from our off-policy data. The effective sample size is only a function of the importance weights $w_i = \frac{\pi_\theta(a_i|x_i)}{\mu(a_i|x_i)}$ where $\pi$ is the policy under consideration (parameterized by $\theta$), assigning a probability to each action, given an input $x_i$, $a_i$ is the logged action and $\mu(a_i|x_i)$ is the probability of $a_i$ given $x_i$ under the logging policy. For the effective sample size we use the form

$$n_{\mathrm{e}} = \frac{(\sum_{i=1}^n w_i)^2}{\sum_{i=1}^n w_i^2}$$

as in Chapter 9 of (Owen, 2013). A small effective sample size makes it difficult to know how good $\pi$ really is. There are two things one can do to improve such situation: make $\mathbb{E}[\mu(a)^{-2}]$ small and make $\pi_\theta(a)$ large. We describe how to achieve this in sections 3.3 and 3.4. Some other diagnostics are covered in the context of section 3.6.

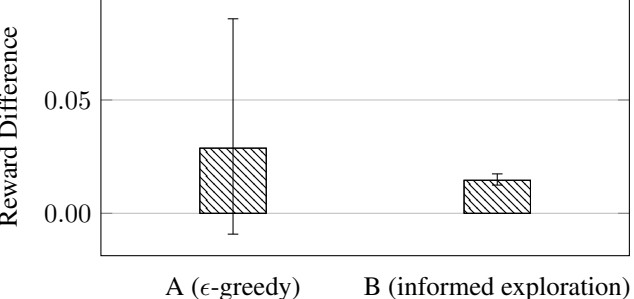

Figure 3. Real data from offline counterfactual policy evaluation results coming from two production systems in the same domain, with 90% bootstrap confidence intervals. **System A** used $\epsilon$-greedy exploration. Refactored **system B** used a more informed exploration approach, producing more actionable estimates. The $y$-axis shows the reward difference from production policy.

## 3.3. Avoid $\epsilon$-greedy

When using logged bandit data, it is important to enable exploration on top of the existing policy. Among exploration strategies, $\epsilon$-greedy is perhaps the simplest. It is also the one that can work with an existing rule-based system. However, $\epsilon$-greedy treats all actions that the underlying policy does not prefer as equally plausible, sampling each of them with an equal probability when it chooses to explore. Furthermore, the choice of whether or not to explore is oblivious to any notion of uncertainty. For example, even with as few as 20 actions and an $\epsilon = 0.1$, the effective sample size of a dataset of size $n$ for a policy that agrees with the logging policy 50% of the time and otherwise acts randomly is $0.01n$. This leads to problems in both training and evaluation as the variance of standard estimators starts dominating (see figure 3.3). When confidence intervals are very large, one can consider evaluation (and even training) with estimators that cap the importance weights and trade some bias for reduced variance. This should be considered a temporary fix and the long-term solution is to have exploration that randomizes among *plausible* alternatives.

Once we realized the seriousness of this problem in our existing system A (cf. Figure 3.3), we invested in engineering work to enable better exploration methods inside our application. However, our current best policy was a mix of rules and ML models. It was created to return a single "best" action, not score all plausible candidates. We did have access to some scores coming from retrieval modules, but each retrieval strategy produced its score on a different scale and none of them was calibrated. Hence, it was hard to use the raw scores in exploration.

Given these constraints, we decided to try a more informed approach. First, we order candidate actions according to a heuristic transformation of their scores, with the action

chosen by the rule-based deterministic policy always at the top position. Then we assigned manually-configured weights $u_k$ that depended only on the initial rank $k$ of each solution. We use $u_1 \geq u_2 \geq \ldots \geq u_j = u_{j+1} = \ldots = u_n$ for some $j$ when there are $n$ actions. The exploration distribution is obtained by normalizing these weights among the available actions (i.e. when three actions are available, the second action is chosen with probability $\frac{u_2}{u_1+u_2+u_3}$). Even this simple and perhaps naïve strategy, which does not incorporate any notion of uncertainty, results in much more robust offline estimates than $\epsilon$-greedy (see **system B** on Figure 3.3).

After promoting the first model-based policy as a default policy, we recommend to discard such heuristics and use the newly promoted policy to drive exploration, e.g. via softmax (Boltzmann) exploration.

### 3.4. Regularize Towards the Logging Policy

When training a new policy, it is important to regularize the policy towards the logging policy. This is definitely not a new piece of advice and is motivated by (Schulman et al., 2015). The rationale is simple: making the model agree more with the logging policy increases the effective sample size, resulting in shorter confidence intervals and reduced overfitting. Empirically, we have found that regularization can increase the effective sample size by 5x (thus reducing the width of the confidence interval by $\sqrt{5}$) without substantial impact on the mean reward. The functional form of the regularizer has not been as important; either direction of KL divergence between new and old policy has worked well and also other kinds of divergences such as total variation distance.

### 3.5. Design an Architecture Suited to RL

Enterprise systems typically use machine learning only inside isolated components in conjunction with a number of rules, controlled by code and manual configuration. It is common to call a model only in limited scenarios or override a model's decision afterwards. For instance, in the Microsoft Virtual Agent's original architecture, numerous manually configured rules took precedence in execution. The intent disambiguation module, which we wanted to optimize, was either called with limited number of candidates, not called at all, or its decision was overridden by downstream components.

We call this type of architecture "rules-driven". It may contain ML components, but they are treated just as utility functions which can be called (or not) whenever needed and their result can be overridden without consequences. This is perhaps a natural way of designing an enterprise system and it may also work well enough with ML components based on supervised learning.

However, applying RL to a rules-driven architecture poses a real challenge. E.g. in our case, rules filtering candidates limited the RL agent's ability to make significant impact to the end-to-end system. In cases when the agent's component was not called at all, we missed valuable logs in our RL platform which we could have used in training or at least in offline analysis. Overriding the policy's decision is perhaps the worst of all, as it introduces additional noise to data logged by the agent (even if you design the system such that the final overridden outcome is logged, it is much harder to find the correct probability of such action).

Of course, one cannot simply replace all the rules with an RL agent backed by a deep learning model – especially in a mature enterprise application. Many of the rules in place are crucial and should not be broken. One of the most important rules in the Microsoft Virtual Agent is that if the user expresses a desire to talk to a human support agent, the systems needs to respect it – no other action is allowed. We call such important regulations "business rules".

On the other hand, most other rules in our system were not really important from a business perspective. They were just assumptions made by developers of the initial version of the application, mainly due to a lack of data which could be used to build a more accurate model. For instance, one such rule in our system was favoring multi-step dialog scripts over support articles. While this is a reasonable assumption in general, it definitely does not apply to all queries and contexts (e.g. we may not have a dialog script troubleshooting some uncommon issues, but it's very likely we have a web article covering it). Without having access to data coming from real traffic with exploration, basing a cold-start policy on some hypothesis is a reasonable practice.

These observations motivated us to refactor the system to a design pattern that we call an "RL-driven architecture" (see Figure 4) where business rules are separated from the policy component. The policy can be populated with cold-start rules at first. The RL agent is allowed to explore around the current policy, but it is never allowed to take an action breaking any of the business rules. In this setting, the RL agent can be called on every request, without the risk of breaking any important business scenarios (like talking to a human agent) and without missing any data in its logs. Only a subset of our legacy rules graduated into business rules, opening up a whole new space of actions for the agent to explore. Finally, the cold-start assumptions can be proven or rejected with real data, and the policy has a chance of improving the end-to-end system in more significant ways which were impossible before. As an additional benefit, an RL-driven architecture brings clarity to the system's implementation, as well as makes it easier to test.

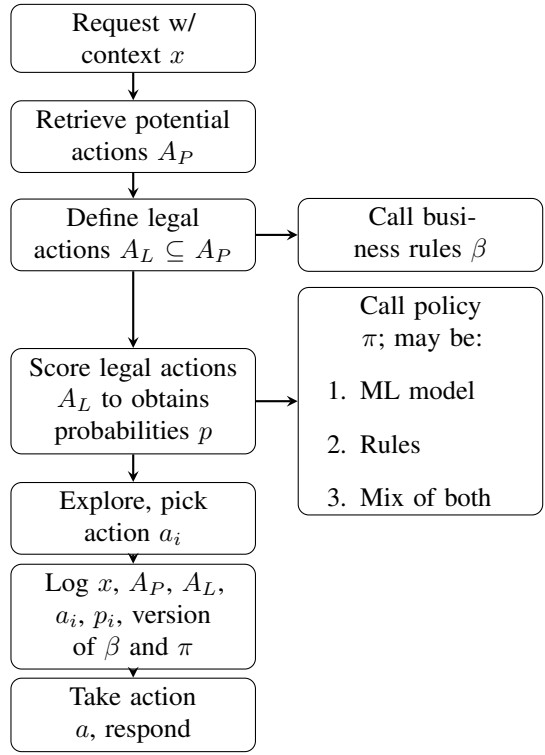

*Figure 4.* RL-driven architecture. All traffic flows through an RL agent. Business rules ensure safety of exploration for every context. Logging not only legal actions $A_L$ but also their superset $A_P$ allows data scientists to reason about changes to business rules in offline analysis.

### 3.6. Balance Randomness with Predictability

Another problem we have encountered when adding exploration to our application is that randomization of the production policy conflicts with typical users' expectation of getting reproducible results. For instance, the user may want to repeat their query for the second time to try a different result from suggested options.

To balance these two conflicting requirements, our sampling code first sets the random seed based on the user's unique identifier concatenated with their query. Similar ideas have been proposed in (Li et al., 2015).

Care needs to be taken to verify that the components that make the random seed are themselves diverse. In a previous iteration, we used a supposedly unique ID from a front-end system. When this ID suddenly started being empty for a large fraction of our data, our offline results no longer matched our online results. Since then we have implemented statistical tests such as those described in (Li et al., 2015) to at least detect when such problems occur.

### 3.7. Consider Starting with Imitation Learning

While it is tempting to work on a policy that can solve the whole task end-to-end, in early stages it is typically more efficient to work on simplified aspects of the problem whose solutions can create stepping stones for addressing the end-to-end task.

For example, it may be worth investing in a model that just *imitates* the production policy on instances where the production policy received large reward. This can be done by simple supervised learning which is much better understood. The resulting policy can still be evaluated using held-out off-policy data and multiple iterations of modeling can be performed until a satisfactory model is obtained. Even if this initial model is only on par with the existing system, it is already a substantial improvement as it can be used for regularizing future policies and more informed exploration.

### 3.8. Consider Simplified Action Spaces

Another possibility for starting simple is to solve certain tasks in isolation. For example, instead of putting together a combinatorial bandit that recommends a whole slate of items in response to a user's query, it might be worth the effort to work only on the decision for the first slot. The first slot offers two distinct advantages: it typically receives more interaction (more people click on it) and counterfactual evaluation is simpler because we do not need to worry about the impact the new policy is having on previous slots.

### 3.9. Don't be Afraid of Principled Exploration

A common worry is metric degradation when exploration is first introduced. While it is certainly possible, and even easy, to degrade business metrics via random and unsafe exploration, in our experience we have found that when exploration is done in a safe and controlled way it is indistinguishable from statistical fluctuations. If the actions already have scores (e.g. from a model learned by supervised learning) these can be used to prioritize the candidates and/or limit the number candidate actions over which we explore. If the actions do not have scores, it might be worth putting together a simple model for the purposes of informed exploration (cf. Section 3.7).

### 3.10. Try to Support Changes in Environment

One of the reasons why reinforcement learning is an appealing alternative to supervised learning is its ability to react to changes in the environment. However, the traditional approach is based just on frequent model updates and letting the agent to adapt to changes step-by-step. Counterfactual policy evaluation cannot predict environment changes.

What we have found though is that in a typical enterprise

application, a large part of the environment is controlled by the application itself: its business rules and various settings controlling these rules (see section 3.5) as well as content that it serves. So even though these are parameters technically outside the RL agent's control, we may still be able to reason about these factors and use tools like counterfactual policy evaluation to manage them in some ways.

The first step we took in this regard was logging not only all legal actions for each context, but also the superset containing top N candidates provided by the retrieval components (see figure 4). Secondly, we decoupled our business rules component such that it could also be evaluated offline based on logged data (similarly to the policy). Now, whenever we plan to introduce some changes to business rules (which happens regularly in a typical enterprise setting), we can run certain validation steps offline. E.g. if a new version of business rules shrinks the number of legal actions in certain contexts, we can use counterfactual evaluation to estimate the impact. If new adjustments expand the space of legal actions, then, thanks to the logged superset of candidate actions, we can at least detect contexts in which expansion happens. We cannot use formal counterfactual evaluation in the latter case but with additional human-labeling we can at least estimate the risk of introducing such changes to business rules. We can also compute offline what probability such new actions would have, according to the current policy, which may further affect our risk assessment.

To manage business rules well, we found it useful to version them clearly and maintain a factory component which is able to produce any desired version. It may be also useful to stamp a trained model with the last version of business rules it has seen in the training data. For example, this may be used to ensure safer deployment of new business rules. At runtime, we can easily spawn two versions of the business rules component (current version and version which the model was trained on). During model inference, we can compare legal actions produced by both versions of business rules. For all actions that were previously illegal and have been legalized by new business rules, it may be safer to cap their probability to some low value. This way, we do not over exploit actions the model did not see in the training data, but allow the system to collect sufficient amount of exploration data. The next version of the model, after seeing enough examples of the newly legalized actions, will be allowed to use them without restrictions.

### 3.11. Cautiously Close the Loop

The great thing about a reinforcement learning system is that you can, in principle, set it and forget it. It can run in a closed training loop automatically on its own past data. In practice however, it may be prudent to at least initially manually close the loop by using a reinforcement learn*ed*

model as one of the treatments in a classic A/B test. This can help iron out potential issues such as a mismatch between offline evaluation and reality. Once there is no cause of concern with single reinforcement learn*ed* models, the next step is to perform an A/B test on the closed loop model updating itself. Such a test can reveal stability of closed-loop dynamics with the model being very different after each update. Regularization towards the logging policy (cf. Section 3.4) can mitigate this issue. Finally, while running the loop as a treatment in an A/B test, it is advised to train only on the portion of the traffic that is actually allocated to the loop treatment, as these will be the conditions that the loop will be operating under when it graduates from the A/B test.

### 3.12. Consider Reward Engineering and Shaping

It is often the case that the reward signal we care most about is the one that is most sparsely provided. In our case, the majority of the users do not respond when asked whether they were able to solve their problem. While it is not clear whether the missing responses are missing at random, we can still try to predict them in a causally sound way. For example, in intent disambiguation we used features such as whether we directly triggered a solution or whether the user clicked the "None of the above" option. An example of a causally dubious feature would be whether the user escalated to a human agent. Even though this would predict the reward well, escalation is thought to be caused by a bad user experience. We then fit a model on held-out data where users have provided responses and use it to impute the missing reward for cases where the users did not provide a response. We have found that training on this imputed reward leads to better results on held out data than training directly on the reward used for evaluation.

### 3.13. Use a Separate Logging Channel

It might be tempting to reuse logging infrastructure that your current application already has, to log your RL agent's decisions with their probabilities. This may be an acceptable short-term solution, but we recommend to quickly separate the logging channel and format used by RL training from the system's typical diagnostic logs. The requirements for these logging channels are usually different. Other developers of the application, who may not be familiar with all the details of the RL infrastructure, may assume that changing some property in the logged data is a low-impact change. However, in case of RL logs, it may impact or completely break model training, correctness of offline counterfactual evaluation etc. Thus, it is advisable to keep it separated.

## 4. Related Work

While there exist many high-quality open source implementations of RL algorithms such as OpenAI baselines (Dhariwal et al., 2017) and dopamine (Castro et al., 2018), our focus is not on setups where the environment is a simulator, but rather the real world. The projects that are close to our goals are the Decision Service (Agarwal et al., 2016), Horizon (Gauci et al., 2018), and RLlib (Liang et al., 2017).

The Decision Service is the basis of our system. It provides a very simple interface for any developer to use RL (Contextual Bandits) by simply specifying features, actions, and rewards. Everything else is handled automatically. Policies are constrained to be expressible by Vowpal Wabbit which is used for training new iterations of the policy. Deployment of new policies happens automatically on a regular schedule, depending on how quickly data becomes out of date (e.g. as fast as every 15 minutes in a news app).

Horizon is based on the same principles as the Decision Service but has added the flexibility of optimizing any kind of ONNX model. It is close to our extension of the DS. To the best of our knowledge, Horizon does not set up a distributed logging service for reliable logging, while the DS uses capabilities built in Azure (Event Hubs).

RLlib has very recently added extensions to support RL from offline data. The original motivation of RLlib was to perform RL in a simulated environment with very easy to use abstractions that can seamlessly scale the computation without the user having to worry about details of distributed systems. We find that RLlib is very focused on training while multiple other components, including logging, exploration, diagnostics, and counterfactual evaluation, need to come together in a real-word RL application.

## 5. Conclusions and Future Work

We have described two applications of single step RL in the Microsoft Virtual Agent along with practical issues that arise in real world Contextual Bandit and RL applications. We hope the practical lessons we have learned in our domain can help other RL practitioners and perhaps even guide RL and systems researchers in the development of better tools for solving real world RL problems.

In the future we will move towards short episodic RL where it is important to do proper credit assignment. For this we plan to rely on a reduction approach (Daumé III et al., 2018) which can operate well with the rest of our existing system.

## Acknowledgements

We thank Markus Cozowicz, Marco Rossi, Rafah Hosn, and John Langford for help with extending the DS system, Brian Bilodeau, and the Customer Care Intelligence Team, for their support with the Intent Disambiguation use case and Mary Buck and the Digital Customer Support team for their help with the Contextual Recommendation scenario.

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
