# OpenReview forum: "Lessons from Contextual Bandit Learning in a Customer Support Bot"
_ICML.cc/2019/Workshop/RL4RealLife — RL4RealLife 2019_

### Official Review · AnonReviewer2 · 2019-05-22
**The paper discusses several practical lessons that the authors learn from the application of RL in Microsoft Virtual Agent, which I believe provides valuable advice for RL practitioners in other domains. The paper is very clear and well aligned with the goal of this workshop.**

**Rating:** 5
**Confidence:** 4

**Review:**

* Quality

Very solid. Provide a lot of insights into how to develop a real-world RL application.  Their rationale of making certain decision during the development process is very helpful.

* Clarity

Clear and well­-written. Most points about the important factors to consider in designing a real-world RL are well explained using the example of Microsoft Virtual Agent.

* Significance

The paper discusses several practical lessons that the authors learn from the application of RL in Microsoft Virtual Agent. These lessons are valuable to RL practitioners in other applications.

---

### Official Review · AnonReviewer1 · 2019-05-24
**Interesting lessons learned from using bandits in the real world (though slightly misleading title)**

**Rating:** 4
**Confidence:** 3

**Review:**

This paper shares some lessons learned from applying contextual bandits to two customer support applications at Microsoft. It describes how bandits are used for the problems of intent disambiguation in a help dialog and suggesting help articles from the settings page based on context. After describing the applications, the paper describes some lessons learned from setting them up, including how to extend existing systems for bandits (which they did largely building on the Decision Service), tracking effective sample size, avoiding pure epsilon-greedy to reduce variance in counterfactual metrics, regularizing towards the logging policy, and how to get bandits to work well with existing business logic.

Overall, this is a well-written paper that describes several practical lessons from getting a bandit system working in a real application. While much of what is in the paper has been published across the literature, the paper does a good job at assembling the many lessons together in a way that would be helpful to people considering using bandits in an application.  As such, the paper should be interesting to practitioners of RL and is clearly within the scope of the workshop.

However, the one thing I would suggest changing in the paper is that the title focuses on “Reinforcement Learning” but the actual examples are just using contextual bandits. While the text generally calls this out, it should be clarified in the title and text that it really is about the bandit use-case and not more broad RL learnings.

Pros:
* Describes a couple real-world applications of a bandit.
* Provides a variety of lessons learned for deploying the bandit applications to the real world.
* Well written and easy to read.

Cons:
* Some of the lessons-learned claims would be better backed up with more data or a clearer explanation of the experiments that led to the lesson. For example, how the non-epsilon greedy explore was done.
* Some aspects like the discussion of using deep learning models, didn’t really provide much detail to actually be useful for other applications.
* The related work section focused mostly on libraries/systems for doing RL, it would be good to see referenced other work on using RL for chatbots and how this compares.

Suggested Revisions:
* Change the title to make clear that the paper is about bandits, not about other forms of RL.
* The first line of the intro seems a bit overblown; consider revising the statement (or backing it up)
* p2: “significant amount of intents” -> “significant number of intents"
* In section 2.2 it would be good to explain why 18 candidates were considered (why not more or less?)
* p3: “could in the future be used for bootstrapping” -> “could be used to bootstrap”

---

### Decision · Program_Chairs · 2019-05-28

Accept